# Discrimination of the Geographical Origin of the Lateral Roots of *Aconitum carmichaelii* Using the Fingerprint, Multicomponent Quantification, and Chemometric Methods

**DOI:** 10.3390/molecules24224124

**Published:** 2019-11-14

**Authors:** Lu-Lin Miao, Qin-Mei Zhou, Cheng Peng, Chun-Wang Meng, Xiao-Ya Wang, Liang Xiong

**Affiliations:** 1School of Pharmacy, Chengdu University of Traditional Chinese Medicine, Chengdu 611137, China; 18408210835@163.com (L.-L.M.); zhqmyx@sina.cn (Q.-M.Z.); mengchunw@126.com (C.-W.M.); 18234447532@163.com (X.-Y.W.); 2State Key Laboratory Breeding Base of Systematic Research, Development and Utilization of Chinese Medicine Resources, Chengdu University of Traditional Chinese Medicine, Chengdu 611137, China; 3Institute of Innovative Medicine Ingredients of Southwest Specialty Medicinal Materials, Chengdu University of Traditional Chinese Medicine, Chengdu 611137, China

**Keywords:** Fuzi, the lateral roots of *Aconitum carmichaelii*, geographical origin, alkaloids, HPLC fingerprint, chemometric analysis

## Abstract

Fuzi is a well-known traditional Chinese medicine developed from the lateral roots of *Aconitum carmichaelii* Debx. It is rich in alkaloids that display a wide variety of bioactivities, and it has a strong cardiotoxicity and neurotoxicity. In order to discriminate the geographical origin and evaluate the quality of this medicine, a method based on high-performance liquid chromatography (HPLC) was developed for multicomponent quantification and chemical fingerprint analysis. The measured results of 32 batches of Fuzi from three different regions were evaluated by chemometric analysis, including similarity analysis (SA), hierarchical cluster analysis (HCA), principal component analysis (PCA), and linear discriminant analysis (LDA). The content of six representative alkaloids of Fuzi (benzoylmesaconine, benzoylhypaconine, benzoylaconine, mesaconitine, hypaconitine, and aconitine) were varied by geographical origin, and the content ratios of the benzoylmesaconine/mesaconitine and diester-type/monoester-type diterpenoid alkaloids may be potential traits for classifying the geographical origin of the medicine. In the HPLC fingerprint similarity analysis, the Fuzi from Jiangyou, Sichuan, was distinguished from the Fuzi from Butuo, Sichuan, and the Fuzi from Yunnan. Based on the HCA and PCA analyses of the content of the six representative alkaloids, all of the batches were classified into two categories, which were closely related to the plants’ geographical origins. The Fuzi samples from Jiangyou were placed into one category, while the Fuzi samples from Butuo and Yunnan were put into another category. The LDA analysis provided an efficient and satisfactory prediction model for differentiating the Fuzi samples from the above-mentioned three geographical origins. Thus, the content of the six representative alkaloids and the fingerprint similarity values were useful markers for differentiating the geographical origin of the Fuzi samples.

## 1. Introduction

*Aconitum carmichaelii* Debx. (Ranunculaceae) is an herbaceous flowering plant mainly distributed in China, Korea, and Japan. The roots were first recorded as a medicine in the Chin bamboo slips of Choujiatai in the Qin Dynasty (221 B.C.–207 B.C.). Since the Han Dynasty (202 B.C.–220 A.D.), the lateral roots and the tap roots of the plant have been used separately. The lateral roots of *A. carmichaelii*, also called “Fuzi”, are extensively used with excellent clinical efficacy; however, the medicine does have a significant toxicity [1,2]. Fuzi is frequently used for the treatment of acute myocardial infarction, coronary heart disease, chronic heart failure, and rheumatic arthralgia. The significant toxicity of Fuzi mainly affects the central nervous system, heart, muscle tissue, and development of embryos [3,4]. Therefore, the analysis of the toxic and active components, and quality control of Fuzi are crucial to its effective use in clinical settings.

Alkaloids are the major type of active substances in Fuzi [5,6]. Among the hundreds of reported alkaloids, aconitine, mesaconitine, hypaconitine, benzoylaconine, benzoylmesaconine, and benzoylhypaconine (Figure 1) have been studied extensively; these are marker components of Fuzi in the Chinese Pharmacopoeia [1] and the local quality standards in China [7,8]. The first three alkaloids are representative diester-type diterpenoid alkaloids. According to modern pharmacological studies, they are the main toxic components of Fuzi, and they have a strong cardiotoxicity and neurotoxicity [9,10,11,12]. Their contents are also strictly limited to less than 0.020% in the Chinese Pharmacopoeia [1]. Benzoylaconine, benzoylmesaconine, and benzoylhypaconine are representative monoester-type diterpenoid alkaloids that possess a lower toxicity [12,13,14,15,16] and more activities [17,18,19] than diester-type diterpenoid alkaloids. Thus, the total content of these three alkaloids should not be less than 0.010%, according to the Chinese Pharmacopoeia [1].

*A. carmichaelii* is mainly distributed in the south of the Yangtze River in China [3]. Sichuan Province and Yunnan Province are two of the main growth areas for the plant [20]. Jiangyou County in Sichuan Province has been cultivating *A. carmichaelii* for more than 1300 years. Traditionally, Fuzi from Jiangyou is considered the best. Because of the increasing demands for the herbal medicine, Butuo County in Sichuan Province, Yulong County and Jianchuan County in Yunnan Province, and other places are cultivating *A. carmichaelii*. Now, the output of Fuzi in Butuo and Yunnan exceeds that in Jiangyou, which accounts for a very large proportion of the market. The geographical origin of herbal medicines is an important factor influencing the quality and price of the medicinal materials [21]. Few researchers have focused on the origin difference analysis of Fuzi based on the content and pharmacological effects of the chemical constituents [22,23,24]. Some researchers have applied chromatographic fingerprints to identify the geographical origin of Fuzi [25]. However, the relevance of the geographical origin and chemical compositions of Fuzi is still unclear, so a combined method of chemical fingerprinting and multicomponent quantification with chemometric analysis is worth developing.

In this study, 32 batches of Fuzi were collected, including 20 batches from Sichuan Province (eight from Jiangyou and twelve from Butuo) and 12 batches from Yunnan Province (eight from Yulong and four from Jianchuan). Reversed-phase HPLC accounts for over 90% of all HPLC separations in several fields, including pharmaceutical chemistry and the analysis of the bioactive compounds of plants [26], so the content determination and fingerprint in this study were analyzed in reversed-phase mode. Cluster analysis, principal component analysis, and linear discriminant analysis were used to analyze the geographical origin differences of Fuzi based on the content of the above-mentioned 6 representative alkaloids (aconitine, mesaconitine, hypaconitine, benzoylaconine, benzoylmesaconine, and benzoylhypaconine).

## 2. Results and Discussion

### 2.1. Optimization of the Extraction Conditions

The extraction method was optimized so as to obtain more information about the chemical constituents of Fuzi. Different extraction solvents (*v*/*v* 1:1 mixed solution of isopropanol-ethyl acetate, dichloromethane, ethyl acetate, and methanol), extraction times (20, 30, and 40 min), and ratios of material to liquid (1:25, 1:35, and 1:50) were investigated. The results showed that the extraction using the isopropanol-ethyl acetate (*v*/*v* 1:1) mixed solution afforded the highest extraction efficiency and the most abundant chromatographic peaks. Ultimately, the best extraction method was the extraction using the isopropanol-ethyl acetate (*v*/*v* 1:1) mixed solution, with a solid–liquid ratio of 1:25 for 30 min.

### 2.2. Optimization of the HPLC Conditions

HPLC with an ultraviolet variable wavelength detector was chosen for the quantification. We compared the chromatograms at 235, 240, 254, and 280 nm with the same sample solution, and found that the alkaloids have a stronger absorption at the 235 nm wavelength. The chromatographic columns, including the Eclipse Plus C_18_ (3.5 μm, 100 × 4.6 mm), Alltima^TM^ C_18_ (5 μm, 250 × 4.6 mm), and Kromasil C_18_ (5 μm, 250 × 4.6 mm), were tested. The Kromasil C_18_ column was the most suitable one. Different mobile phases, including the acetonitrile/tetrahydrofuran (*v*/*v* 25:15)–0.1 mol/L ammonium acetate aqueous solution, acetonitrile–0.2% acetic acid aqueous solution, methanol–0.1% diethylamine aqueous solution, acetonitrile–0.1% diethylamine aqueous solution, methanol–0.1% ammonia aqueous solution, and acetonitrile–0.1% ammonia aqueous solution, were examined. The results showed that the acetonitrile/tetrahydrofuran (*v*/*v* 25:15)–0.1 mol/L ammonium acetate aqueous solution was the most appropriate. In order to achieve a good separation of analytes, the following eluting gradients of the mobile phases (shown in Table 1) were tested. Eluting Gradient 4 provided the best separation efficiency out of all of the eluting gradients. In addition, analyses of the separation effects of the column temperature (20, 25, and 30 °C) and injection volume (5, 10, and 15 μL) indicated that a low temperature and excessive injection volume may lead to a delay of the peak retention time and peak tailing. Consequently, according to the peak retention time, peak shapes, and peak resolutions, the optimum HPLC condition was that the separation was performed on a Kromasil C_18_ column (5 μm, 250 × 4.6 mm) using a mixture of acetonitrile/tetrahydrofuran (*v*/*v* 25:15; A) and 0.1 mol/L ammonium acetate aqueous solution (B) as the mobile phase; eluting Gradient 4 (Table 1) was applied in the gradient elution; the flow rate was adjusted to 1 mL/min; the column was kept at 30 °C; the quantification wavelength was set at 235 nm; and the injection volume was 5 μL. Brief results of optimization process are presented in Appendix A.

### 2.3. Method Validation: Precision, Linearity, Accuracy, Specificity, Stability, and Ruggedness

The calibration curves of the six alkaloids showed a good linearity based on the regression analysis within the tested ranges (r^2^ > 0.9995; Table 2). The relative standard deviation (RSD) values of the precision are in the range of 1.49% to 3.16%, and the RSD values of the repeatability range from 1.53% to 4.70%. As for the stability, the six standards showed a good stability in 24 h at room temperature, with RSDs less than 3.94%. The average recovery varies between 98.81% and 101.68% with the RSDs less than 4.57%, which indicated that the established method was suitable for the quantitative analysis of the six compounds.

### 2.4. Quantitative Analysis of the Six Alkaloids in the 32 Samples

The optimized chromatographic method was applied to the quantitative analysis. The content of each alkaloid was calculated according to their respective calibration curves (Appendix A), and the statistical analysis of results are shown in Figure 2. The Fuzi from Jiangyou has the highest content of hypaconitine, and the lowest content of mesaconitine and aconitine. On the contrary, the Fuzi from Yunnan has the lowest content of hypaconitine and the highest content of mesaconitine and aconitine. Notably, the total content of the above three diester-type diterpenoid alkaloids in the Fuzi from Jiangyou (0.1226% ± 0.0126%) is much lower than that from Butuo (0.1618% ± 0.0478%) and Yunnan (0.1819% ± 0.0534%). In addition, the content of the monoester-type diterpenoid alkaloids represented by benzoylhypaconine, benzoylmesaconine, and benzoylaconine also varies according to the geographical origin (Figure 2E–G). It should be noted that the amounts of benzoylaconine and benzoylhypaconine were very low, and could not be quantified or detected in the Fuzi samples from Butuo and Yunnan, which is different from the Fuzi samples from Jiangyou. The content of benzoylmesaconine in the Fuzi from Butuo and Yunnan is lower than that in the Fuzi from Jiangyou. Overall, the Fuzi from Jiangyou presented a higher level of monoester-type alkaloids and a lower level of diester-type alkaloids. According to the recent pharmacological and toxicological studies of Fuzi’s monoester- and diester-type alkaloids, these results probably mean that the Fuzi from Jiangyou has a high efficacy and low toxicity, which is consistent with the traditional view—“Fuzi from Jiangyou is the best”.

Figure 2 shows that mesaconitine has a higher relative content than aconitine and hypaconitine in most of the Fuzi samples. Therefore, taking mesaconitine as a reference compound, the content ratios of benzoylmesaconine/mesaconitine, hypaconitine/mesaconitine, and aconitine/mesaconitine of the 32 samples were calculated; the results are shown in Figure 3. It is worth noting that the content ratios of benzoylmesaconine/mesaconitine (Figure 3A) are closely related to the geographical origin. The ratio values of the Fuzi samples from Butuo and Yunnan are 0.164–0.222 and 0.110–0.222, and the average ratios of the samples from the regions are 0.198 ± 0.019 and 0.150 ± 0.029, respectively. However, the values of the Fuzi samples from Jiangyou range from 0.670 to 0.815, with an average of 0.720 ± 0.049, more than three times the value of the Fuzi from the other two origins. In addition, the ratio of diester-type/monoester-type alkaloids also showed the characteristic of Fuzi from Jiangyou (Figure 3D). Statistical analysis indicated that there was significant difference between the ratio of Fuzi from Jiangyou and the ratios of Fuzi from the other two origins. Thus, the content ratios of benzoylmesaconine/mesaconitine and diester-type/monoester-type alkaloids may be chemical features that can be used to identify the geographical origin of Fuzi. However, more samples from other origins are needed for further validation. In addition, the content ratios of hypaconitine/mesaconitine and aconitine/mesaconitine were not available for discriminating the geographical origin of Fuzi, because of the instability of the ratios in Fuzi from Jiangyou, although they are generally higher than those in the Fuzi from the other two origins.

### 2.5. HPLC Fingerprint and SA (Similarity Analysis)

The HPLC chromatograms of the 32 samples were exported as CSV files, and imported into the fingerprint system for further analysis. Six common peaks were selected in the chromatograms, and a total fingerprint chromatogram of 32 batches and a reference fingerprint chromatogram were produced (Figure 4A,B, respectively). A similarity analysis was performed using the system, based on the correlation coefficients of the raw data. The similarity value of each chromatogram of all of the samples to the reference fingerprint chromatogram ranged from 0.637–0.992 (Table 3). The similarity values of the chromatograms of the eight batches of Fuzi from Jiangyou to the reference fingerprint chromatogram were relatively low, while the values of all of the batches of Fuzi from Butuo and Yunnan were above 0.9. The overall characteristics of the Fuzi samples from Jiangyou were very different from those of the Fuzi samples from the other two geographical origins.

### 2.6. HCA (Hierarchical Cluster Analysis)

HCA can show the degree of correlation between a large number of samples. The similarity or dissimilarity between all of the samples is usually represented by a tree diagram [27]. The content of the six alkaloids of the 32 batches of Fuzi were used to form a 6 × 32 data matrix. The data matrix was imported into SPSS 20.0 software. The average linkage between the groups and the square Euclidean distance were used as the standard interval for the cluster analysis. All 32 samples were divided into four major clusters (A1, A2, B1, and B2) when the squared Euclidean distance was 7.5 (Figure 5). The Fuzi from Yunnan (YN) and Butuo (BT) were characteristically closed (A1 and A2). YN-1–YN-7 and YN-9–YN-11, together with BT-7, BT-10, and BT-11, were partitioned as cluster A1. They all contained higher amounts of diester-type diterpenoid alkaloids than A2, which consisted of the remaining batches of Fuzi from Yunnan and Butuo. The samples from Jiangyou were separated into B1 and B2.

### 2.7. PCA (Principle Component Analysis)

PCA combines multiple variables into a small number of variables that reflects the overall information through linear transformation. PCA can retain the main information of a large number of variables, and reduce the dimension of the data. It is a common multivariate data analysis method [27,28]. The same 6 × 32 data matrix was applied to the PCA analysis. The results of the variance showed that the eigenvalues of the first two components were 3.150 and 1.973, greater than 1, and their contribution rates to the total variance were 52.5% and 32.9%. Therefore, two principal component factors (PC1 and PC2) were extracted for calculation, which contained the most information on the raw data. R2 and Q2 were used to assess the obtained model performances, and the two statistical parameters were 0.854 and 0.589, respectively. The biplot of the PCA is shown as Figure 6A. A two-dimensional projection made by the comprehensive principal component factor scores of each batch of Fuzi is shown in Figure 6B. It shows that the Fuzi from Jiangyou (JY) and the Fuzi from the other two origins (BT and YN) are obviously divided into two main categories. The Fuzi from Butuo and Yunnan (BT and YN, respectively) were further clustered into two parts. Only two Fuzi samples from Yunnan, YN-8 and YN-12, were similar to the Fuzi from Butuo.

### 2.8. LDA (Linear Discrimination Analysis)

HCA and PCA are unsupervised methods without classification labels. LDA builds a model based on a data matrix and the known classified information [29]. According to the classification results of the HCA and PCA, the Fuzi from Jiangyou differs from the Fuzi from Butuo and Yunnan. Although the Fuzi samples from Yunnan and Butuo have some similarities in the analysis results, most of the batches are classified by origin. Therefore, we defined the Fuzi from the three origins as three groups. An analysis of variance (ANOVA) between the defined three groups were employed for the statistical significance testing of the PCA scores. The PC1 scores of three groups had a significant difference between each other (*p* < 0.01), and the PC2 scores of the Fuzi from Jiangyou and the Fuzi from Butuo also showed a difference (*p* < 0.05). Then, the PCA scores were used as the input data for LDA in order to calculate the canonical factor. The unsupervised PCA followed by supervised LDA was found to reduce the chance of over-fitting that may occur with a pure LDA model [30]. Two canonical discriminant functions were obtained, Functions 1 and 2, containing 99.4% and 0.6% of the raw data, respectively. The scores for Functions 1 and 2 of each sample are shown in Figure 6C. Leave-one-out cross-validation showed that the classification accuracy of the discriminant function is 90.6%. The cross-validated specificity values of JY, BT, and YN are 100%, 94.7%, and 90.5%, respectively, and the sensitivity values of JY, BT, and YN are 100%, 91.7%, and 90.9%, respectively. Fisher’s discriminant functions of the Jiangyou, Butuo, and Yunnan groups are shown in Table 4. As shown in Figure 6C, LDA confirms the results of PCA, and provides a simple and effective model to distinguish the geographical origin of Fuzi.

All of the above three analytical methods could be applied to discriminate the geographical origin of the Fuzi samples. Most of the samples were classified according to their geographical origin. The Fuzi samples from Jiangyou were quite different from the Fuzi samples from Butuo and Yunnan. This significant difference may be related to the planting environment and climate, planting techniques, and harvesting time. As shown in Figure 7, Jiangyou is located in the northwest part of the Sichuan Basin. The altitude of the planting base is about 530 m above sea level. The climate of this area has a hot and long summer, warm winter, abundant precipitation, and a synchronization of the rain and heat. However, Butuo in Sichuan Province and Yulong and Jianchuan in Yunnan Province are situated in the Yunnan-Kweichow Plateau, where the altitude is 2000 m above sea level. These regions belong to the plateau climate area, where the climate is characterized by small differences per year, but large differences per day. In addition, it is warm in the winter and cool in the summer [31]. Environment has an important impact on the morphology and physiology of plants, and it is one of the factors that determines the quality of medicinal materials [32]. Altitude differences usually lead to changes in environmental factors, such as temperature, precipitation, light, and soil, which make the environmental conditions of plant growth more complex, and also cause complex adaptability changes of plants to the environment [32]. Thus, it can be speculated that the planting environment and climate are the main reasons for the differences in the Fuzi samples from Jiangyou, Butuo, and Yunnan.

During the plant’s long cultivation history, a traditional pattern of breeding the roots of *A. carmichaelii* on high mountains, and then planting the plant on the plains has developed in Jiangyou. After a series of characteristic cultivation processes, like removing the top part of the plant and the redundant lateral roots, only two–three large lateral roots are retained and harvested in July [33,34]. Butuo in Sichuan Province is a mountainous area with a high altitude, and the cultivation history of Fuzi there is short—it has only been grown for about 30 years. *A. carmichaelii* planted in this area does not undergo the above-mentioned cultivation process. In addition, because of the cold climate, the harvest period is in September [35]. The cultivation area of *A. carmichaelii* in Yunnan Province belongs to the plateau climate zone. The planting pattern is similar to that found in Butuo, and the plant is harvested in October. This process may contribute to the similarity of Fuzi samples from Butuo and Yunnan.

## 3. Materials and Methods

### 3.1. Plant Material

A total of 32 batches samples were collected in 2017, of which eight batches were from Planting Base of Huarun Sanjiu (Ya’an) Pharmaceutical Co., Ltd. in the Villages of Puzhao, Shuangsheng, Qiaolou, Zhulin, Yueai, and Longmen in Jiangyou County, Sichuan Province; 12 batches were from Villages Ruopu, Taijinai, and Luo’en in Butuo County, Sichuan Province; and 12 batches were from Villages Taiping and Hongyan in Yulong County and Hongxing Village in Jianchuan County, Yunnan Province. The plant identities were verified by Associate Professor Ji-Hai Gao (Chengdu University of TCM, Sichuan, China).

### 3.2. Reagents and Chemicals

Mesaconitine (110799-200404) and hypaconitine (110798-200404) were purchased from the National Institute for Control of Biological and Pharmaceutical Products of China (Beijing, China). Aconitine (MUST-17022206), benzoylaconine (MUST-15012215), benzoylhypaconine (MUST-15010806), and benzoylmesaconine (MUST-15012216) were purchased from Chengdu Must Bio-technology Co., Ltd. (Chengdu, China).

Acetonitrile and methanol of HPLC grade were obtained from Sigma-Aldrich (St. Louis, MO, USA); tetrahydrofuran, diethylamine, ammonia solution, and acetic acid of HPLC grade were purchased from Chengdu Kelong Chemicals Co., Ltd. (Chengdu, China); and ammonium acetate was purchased from Tianjin Kermel Chemicals Co., Ltd. (Tianjin, China). HPLC-grade water was prepared with a Millipore Milli-Q system (Bedford, MA, USA). The other reagents and chemicals of analytical grade were from Chengdu Kelong Chemicals Co., Ltd. (Chengdu, China).

### 3.3. Preparation of Standard Solutions and Sample Solutions

The reference standards were accurately weighed and dissolved in isopropanol-ethyl acetate (*v*/*v* 1:1). Final standard solutions were obtained by diluting to the gradient concentrations with the appropriate amount of isopropanol-ethyl acetate and then stored at 4 °C.

The lateral roots of *A. carmichaelii* were washed, sliced, and dried at 40 °C. The slices were ground into a powder that was capable of passing through a 50-mesh sieve. About 2 g powder was weighed and transferred to a conical flask with a cover; then 3 mL ammonia solution was added. Then, 50 mL isopropanol–ethyl acetate (*v*/*v* 1:1) was added, and the conical flask was weighed with a precision of ±0.01 g. The solution was ultrasonicated for 30 min at a temperature no more than 25 °C. After being cooled to room temperature, the isopropanol–ethyl acetate (*v*/*v* 1:1) was added to compensate for the lost weight. Thirty-five mL of the filtrate was measured and concentrated in vacuo under 40 °C. The residue was dissolved in 4 mL isopropanol–dichloromethane (*v*/*v* 1:1) and filtered through a 0.22-μm nylon filter membrane for the HPLC analysis.

### 3.4. Instrument and Chromatographic Conditions

A chromatography analysis was carried out on an Agilent series 1100 HPLC liquid chromatograph with a quaternary pump, a degasser, an autosampler, a column compartment, and a UV-variable wavelength detector (Agilent Technologies, Palo Alto, CA, USA). The separation was performed on a Kromasil C_18_ column (5 μm, 250 × 4.6 mm) at 30 °C with a flow rate of 1 mL/min of the mobile phase, which consisted of acetonitrile/tetrahydrofuran (*v*/*v* 25:15 A) and the 0.1 mol/L ammonium acetate aqueous solution (B). The mobile phase used as the gradient elution of 15–16% A in 0–14 min, 16–18% A in 14–65 min, 18–20% A in 65–90 min, and 20% A in 90–95 min. The detection wavelength was kept at 235 nm and the injection volume was 5 μL.

### 3.5. Method Validation: Linearity, Precision, Stability, Repeatability, and Recovery

Each concentration of the standard solutions was analyzed, and six calibration curves were constructed with the peak areas and the content of the standard in the injection solution. The precision of the instrument was tested by repeatedly injecting the same sample solutions six times. Six different solutions made from the same sample were used to evaluate the repeatability of the method. The stability was assessed by the same sample solution at 0, 5, 10, 14, 20, and 24 h. The precision, repeatability, and stability were all evaluated based on the RSDs of the peak areas. The recovery reflected the accuracy of the method, which was measured by comparing the determined content and nominal content.

### 3.6. Data Analysis

A SA was carried out on a similarity evaluation system for the chromatographic fingerprint of Traditional Chinese Medicine 2012 vision (Chines Pharmacopoeia Commission, Beijing, China). The ANOVA, HCA, and LDA were analyzed using SPSS 20.0 (SPSS Statistics, Chicago, IL, USA). PCA were analyzed using SIMCA-P 13.0 (Umetrics, Umea, Sweden). GraphPad Prism 5 (GraphPad, CA, USA) was applied on the content comparison of 32 batches of Fuzi.

## 4. Conclusions

In this study, 32 batches of Fuzi from the main producing areas in China were collected for quality evaluation. A new HPLC method was established for the quick and accurate determination of the alkaloids. The representative active diterpenoid alkaloids were measured, and the statistical analysis was performed. The content of the six alkaloids showed differences in the Fuzi samples from the three geographical origins (Jiangyou, Butuo, and Yunnan). Notably, the content ratios of benzoylmesaconine/mesaconitine and diester-type/monoester-type diterpenoid alkaloids may be two chemical features that can be used to differentiate the origin of Fuzi. HPLC fingerprint chromatography was used to reflect the overall phytochemical characteristics of the Fuzi samples, and to evaluate their quality, HPLC fingerprint chromatography was created. The similarity analysis results were calculated with the six common characteristic peaks, and the similarity values of the 32 batches ranged from 0.637 to 0.992, depending on the region. The overall characteristics of the Fuzi samples from Jiangyou were very different from those of the Fuzi samples from Butuo and Yunnan. A further evaluation was performed using chemometric methods (PCA and HCA), and the 32 samples were classified into four groups. The Fuzi from Jiangyou was easily distinguished from the samples from other origins. LDA is a high resolution and satisfactory prediction model for discriminating the geographical origins of Fuzi. It is conjectured that the main reasons for this significant difference of the Fuzi samples may be related to the planting environment, climate, and techniques. In the future, more Fuzi growing regions should be investigated, and molecular biology and epigenetic methods should be used to evaluate the relationship between the medicinal materials and geographical origins.

## Figures and Tables

**Figure 1 molecules-24-04124-f001:**
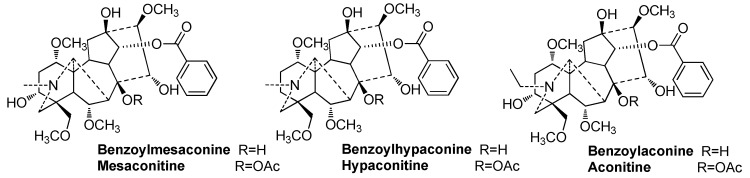
Structures of six representative alkaloids of Fuzi.

**Figure 2 molecules-24-04124-f002:**
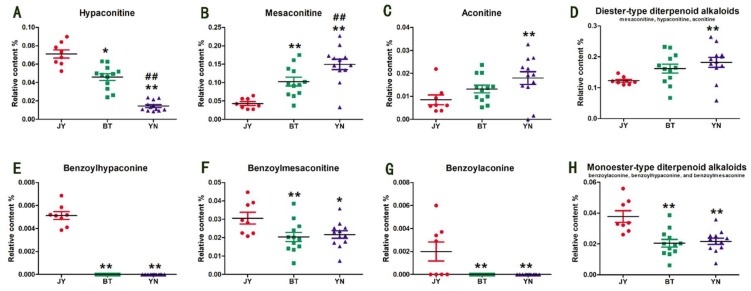
Relative contents of hypaconine (**A**), mesaconitine (**B**), aconitine (**C**), diester-type diterpenoid alkaloids (**D**), benzoylhypaconine (**E**), benzoylmesaconine (**F**), benzoylaconine (**G**), and monoester-type diterpenoid alkaloids (**H**) in the Fuzi samples from Jiangyou, Butuo, and Yunnan. JY is the Fuzi from Jiangyou; BT is the Fuzi from Butuo; and YN is the Fuzi from Yunnan. * *p* < 0.05 vs. JY, ** *p* < 0.01 vs. JY; ^#^
*p* < 0.05 vs. BT, ^##^
*p* < 0.01 vs. BT.

**Figure 3 molecules-24-04124-f003:**
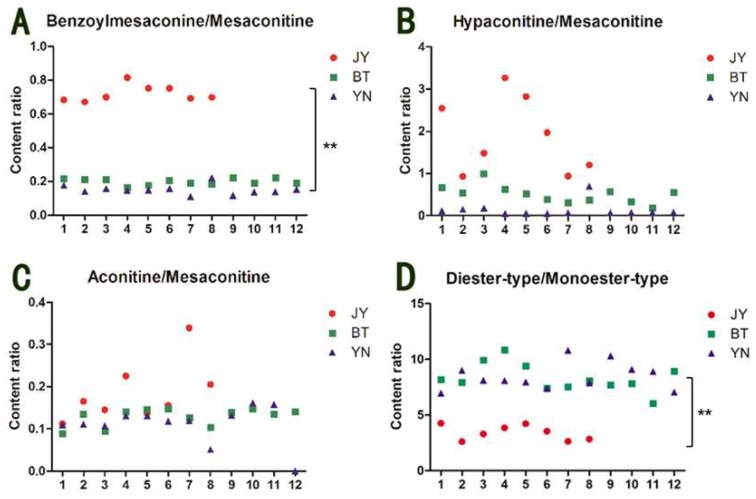
Content ratios of benzoylmesaconine/mesaconitine (**A**), hypaconitine/mesaconitine (**B**), aconitine/mesaconitine (**C**), and diester-type/monoester-type diterpenoid alkaloids (**D**) in the Fuzi samples from Jiangyou, Butuo, and Yunnan. JY is the Fuzi from Jiangyou; BT is the Fuzi from Butuo; and YN is the Fuzi from Yunnan. ** *p* < 0.01 vs. JY.

**Figure 4 molecules-24-04124-f004:**
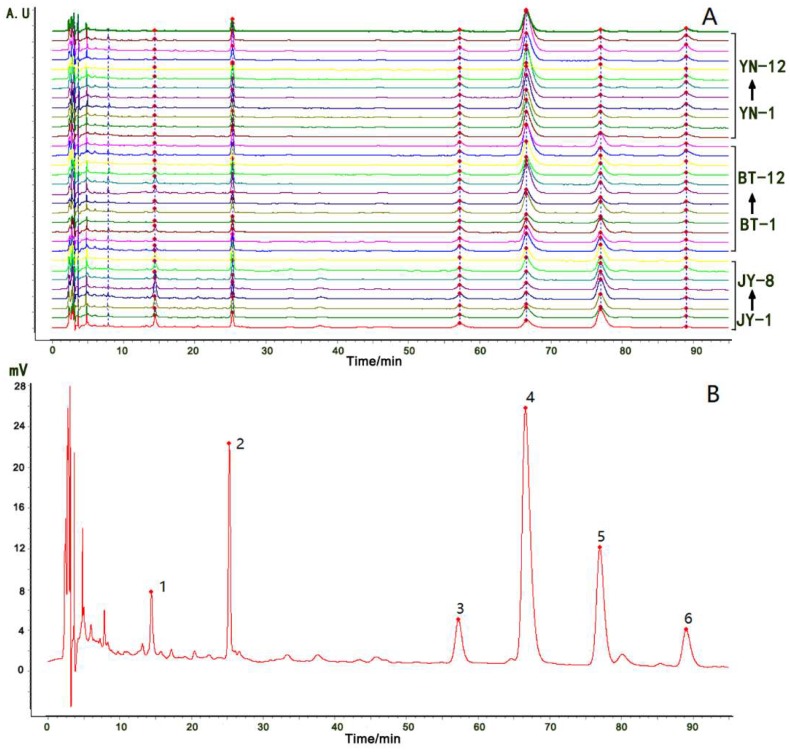
A total fingerprint chromatogram of the 32 batches of Fuzi from Jiangyou, Butuo, and Yunnan (**A**); a reference fingerprint chromatogram of the 32 batches of Fuzi (**B**); Compound 2 is benzoylmesaconine; compound 4 is hypaconitine; compound 5 is mesaconitine; compound 6 is aconitine.

**Figure 5 molecules-24-04124-f005:**
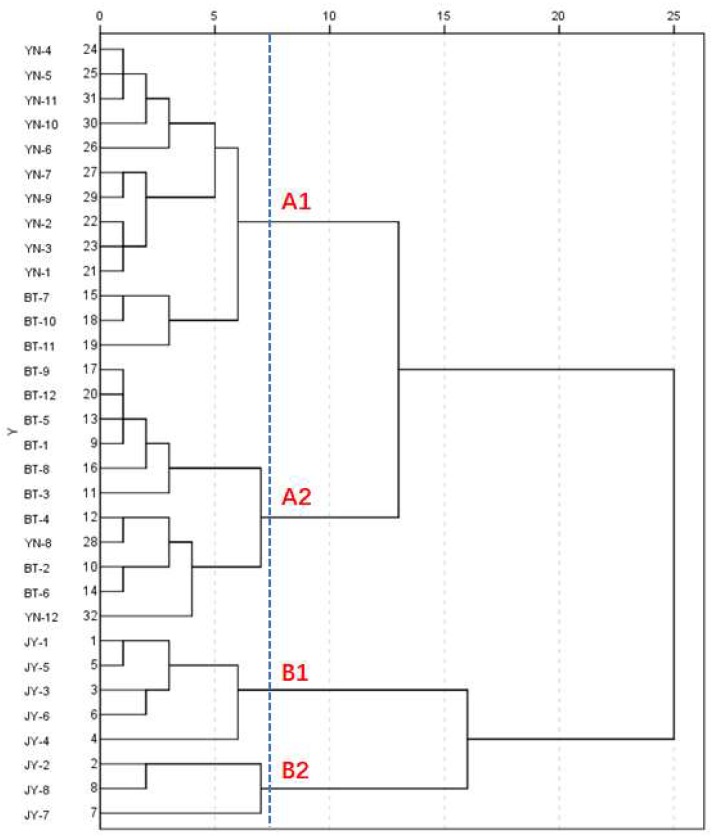
Dendrograms of the hierarchical cluster analysis for the 32 samples. JY is the Fuzi from Jiangyou; BT is the Fuzi from Butuo; and YN is the Fuzi from Yunnan.

**Figure 6 molecules-24-04124-f006:**
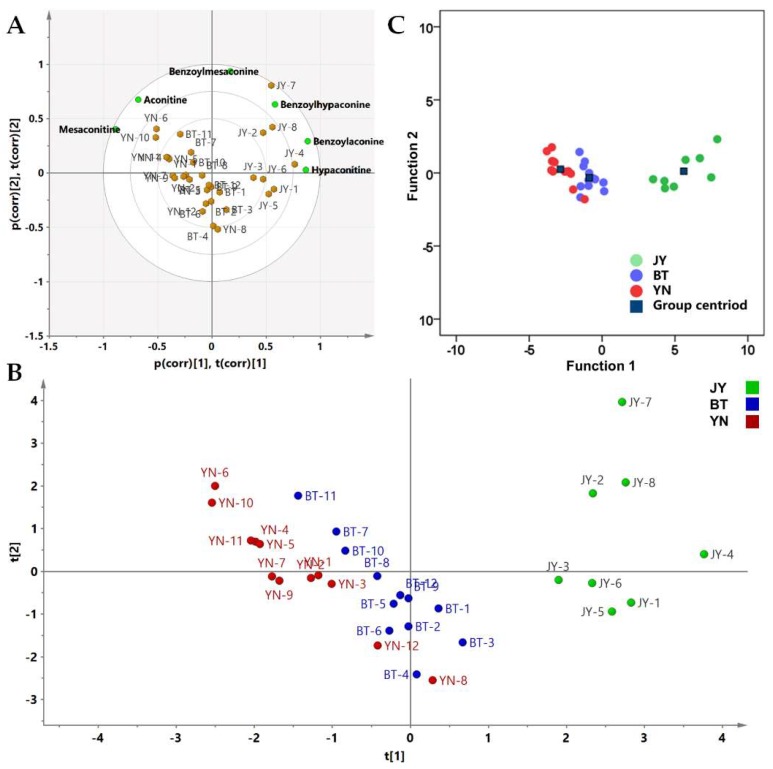
(**A**) Biplot of the PCA; (**B**) 2-D loadings plot and scores of the PCA; (**C**) Scores plot of the canonical discriminant functions of the 32 samples. JY is the Fuzi from Jiangyou; BT is the Fuzi from Butuo; and YN is the Fuzi from Yunnan.

**Figure 7 molecules-24-04124-f007:**
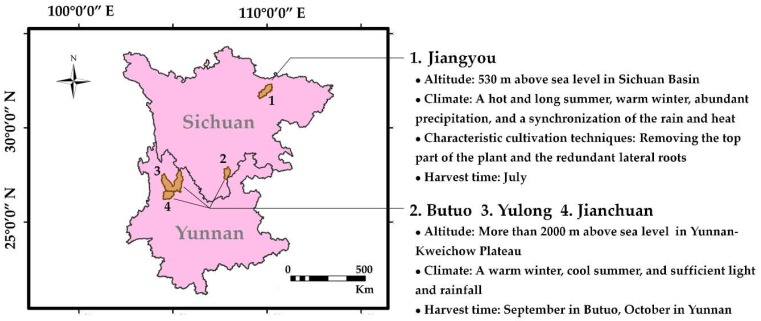
The main cultivated regions of Fuzi in Sichuan and Yunnan Provinces.

**Table 1 molecules-24-04124-t001:** Four eluting gradients examined in optimization of the HPLC conditions of Fuzi.

No.	1	2	3	4
**Eluting Gradient**	15–26% A in 0–48 min, 26–35% A in 48–49 min, 35% A in 49–58 min, 35–15% A in 58–65 min	15–18% A in 0–14 min, 18–20% A in 14–30 min, 20% A in 30–38 min, 20–25% A in 38–40 min, 25% A in 40–48 min, 25–22% A in 48–55 min	15–18% A in 0–14 min, 18–20% A in 14–30 min, 20% A in 30–55 min, 20–25% A in 55–75 min	15–16% A in 0–14 min, 16–18% A in 14–65 min, 18–20% A in 65–90 min, 20% A in 90–95 min

**Table 2 molecules-24-04124-t002:** Regression equation, precision, repeatability, stability, and recovery of the six alkaloids.

Reference Substance	Regression Equation	r^2^	Linear Range/μg	Precision RSD%	Repeatability RSD%	Stability RSD%	Recovery	
							%	RSD%
Benzoylmesaconine	Y = 973.8 X + 33.044	0.9996	0.105–1.05	2.13	1.53	2.04	98.94	4.28
Benzoylhypaconine	Y = 1213.9 X + 17.835	0.9996	0.06–1.2	2.92	4.70	3.35	100.89	4.24
Benzoylaconine	Y = 1020.0 X + 6.894	0.9997	0.035–0.7	3.16	3.88	3.63	101.68	4.57
Mesaconitine	Y = 1042.3 X + 90.868	0.9999	0.2275–4.55	2.14	3.13	2.99	100.01	3.45
Hypaconitine	Y = 1199.3 X + 8.4619	1	0.0988–1.975	1.49	2.31	1.24	100.86	4.16
Aconitine	Y = 1151.7 X + 8.3868	0.9998	0.0355–1.775	3.06	2.60	3.94	98.81	4.50

**Table 3 molecules-24-04124-t003:** Similarity values of the HPLC chromatograms to the reference fingerprint.

Sample	Similarity	Sample	Similarity	Sample	Similarity	Sample	Similarity
JY-1	0.722	BT-1	0.955	BT-9	0.985	YN-5	0.936
JY-2	0.885	BT-2	0.985	BT-10	0.980	YN-6	0.929
JY-3	0.637	BT-3	0.914	BT-11	0.964	YN-7	0.937
JY-4	0.657	BT-4	0.954	BT-12	0.985	YN-8	0.900
JY-5	0.703	BT-5	0.988	YN-1	0.950	YN-9	0.942
JY-6	0.763	BT-6	0.990	YN-2	0.966	YN-10	0.937
JY-7	0.900	BT-7	0.991	YN-3	0.963	YN-11	0.943
JY-8	0.866	BT-8	0.992	YN-4	0.930	YN-12	0.944

**Table 4 molecules-24-04124-t004:** Canonical discriminant functions and Fisher’s discriminant functions of the three groups.

	Canonical Discriminant Functions		Fisher’s Discriminant Functions		
	1	2	JY	BT	YN
PC1	1.917	−0.153	10.729	−1.637	−5.516
PC2	0.682	0.687	3.901	−0.829	−1.771
Constant	0	0	−16.812	−1.542	−5.210
Functions	Function 1 = 1.917 PC1 + 0.682 PC2 Function 2 = −0.153 PC1 + 0.687 PC2		JY = 10.729 PC1 + 3.901 PC2 − 16.812 BT = −1.637 PC1 − 0.829 PC2 − 1.542 YN = −5.516 PC1 − 1.771 PC2 − 5.210		

JY is the Fuzi from Jiangyou; BT is the Fuzi from Butuo; and YN is the Fuzi from Yunnan.

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
