# Peer review of "Discrimination of the Geographical Origin of the Lateral Roots of Aconitum carmichaelii Using the Fingerprint, Multicomponent Quantification, and Chemometric Methods"

_molecules, 2019, doi:10.3390/molecules24224124_

Round 1

Reviewer 1 Report

A very similar work entitled “Comprehensive quality evaluation of the lateral root of Aconitum carmichaelii Debx. (Fuzi): Simultaneous determination of nine alkaloids and chemical fingerprinting coupled with chemometric analysis” (DOI: 10.1002/jssc.201800937) was recently published. In this publication the same plant, the same standards, similar methodology were taken into account. The chemometric techniques are also mostly the same. Therefore the lack of novelty is observed in this work.

Author Response

Comment: A very similar work entitled “Comprehensive quality evaluation of the lateral root of Aconitum carmichaelii Debx. (Fuzi): Simultaneous determination of nine alkaloids and chemical fingerprinting coupled with chemometric analysis” (DOI: 10.1002/jssc.201800937) was recently published. In this publication the same plant, the same standards, similar methodology were taken into account. The chemometric techniques are also mostly the same. Therefore the lack of novelty is observed in this work.

Response: We have read the recently published work and find that there are certain similarities between our article and the published paper. However, there are several differences between the two papers, including methods, results, and discussion. The main differences are as follows.

1) In addition to comparison of the content of representative alkaloids in Fuzi from different origins, we tried our best to explore some chemical markers or features for differentiating the origin of Fuzi. According to the result analysis, we put forward a new idea that some content ratios of the representative alkaloids could be used as chemical features to differentiate the origin of Fuzi. We reported two chemical features in this paper, the ratio of diester-type/monoester-type alkaloids and the ratio of benzoylmesaconine/mesaconitine.

2) The extraction conditions and HPLC conditions are different between two papers. We established the extraction method and the chromatographic process based on the optimal design. 3) The content of some alkaloids is different. For example, we have detected benzoylmesaconine in Fuzi from Butuo, and its content is similar to that in fuzi from Yunnan. However, benzoylmesaconine was not detected in Fuzi from Butuo in the published paper.

In addition, in the HPLC fingerprint, the similarity values are different between two papers. The published paper reported the Fuzi from Butuo and Jiangyou have high similarity (similarity values › 0.8), while we found that the overall fingerprint characteristics of the Fuzi samples from Jiangyou are very different from those of the Fuzi samples from Butuo.

In our paper, possible reasons for these differences were discussed. Although both Butuo and Jiangyou belong to Sichuan Province, the environment and climate of Butuo and Jiangyou are very different. Jiangyou is located in the northwest part of the Sichuan Basin. The altitude of the planting base is about 530 meters above sea level. The climate of this area has a hot and long summer, warm winter, abundant precipitation, and a synchronization of the rain and heat. However, Butuo in Sichuan Province is situated in the Yunnan-Kweichow Plateau, of where the altitude is 2000 meters above sea level. The region belongs to the plateau climate area, where the climate is characterized by small differences per year but large differences per day. What’s more, the planting techniques and harvesting time of Fuzi are very different between Butuo and Jiangyou.

4) Instead of PLS-DA and OPLS-DA in the published work, we used LDA to establish the discrimination model. The discrimination analysis of the two works both provided an efficient prediction model for differentiating the Fuzi of Jiangyou from Fuzi of other origins. The effect of geographical origin on Fuzi was confirmed.

5) Since environment has an important impact on the morphology and physiology of plants, and it is one of the factors that determines the quality of medicinal materials, we focused on the discussion of the environmental and climatic reasons for the differences of Fuzi from different origins. We speculated that the planting environment and climate are the main reasons for the differences in the Fuzi samples from Jiangyou, Butuo, and Yunnan. In addition, we also put forward a point that different planting techniques of Fuzi in different origins may have a great influence on the chemical components of Fuzi. In our follow-up study, we will perform a planting experiment combined with the chemical analysis to confirm this deduction.

Reviewer 2 Report

The authors (Miao et al.) present an study on discrimination of the geographical origin of the lateral roots of Aconitum Carmichaelli using fingerprint RP-HPLC and chemometrics. The research work is interesting and would appeal to the broad readership of Molecules. However, several important points need to be addressed before it can be accepted for publication.

In the Introduction section the authors should further elaborate why RP-HPLC was utilized to analyze Fuzi extracts. Please also make clear (throughout the manuscript) that the separation mode was reversed-phase. A recently published review notes that the RP mode accounts for > 90 % of all HPLC separations in several fields including pharmaceutical chemistry, and analysis of (biologically)-active compounds of plant origin. (10.1021/acs.chemrev.8b00246) What was the rationale of choosing Kromasil C18 as the optimal column for the RP-HPLC separation? Were chromatographic tests performed (to determine e.g., hydrophobicity, silanol activity, steric selectivity)? If yes, please elaborate more on the optimization of RP-HPLC conditions in the manuscript. The authors are referred to the previously-mentioned review article for information on RP-HPLC column selection (10.1021/acs.chemrev.8b00246). Instead of performing the PCA&LDA analyses separately, the authors may want to consider coupling PCA (or even PLS) to LDA, whereas the PCA scores (computed from the chromatographic data) would be used as input to LDA in order to calculate the canonical factors instead of chromatographic data. In such a manner the noise levels would be minimized. The models lack comprehensive validation. Since the number of samples is quite low, please employ leave-one-out cross-validation (D. L. Massart, et al. Handbook of Chemometrics and Qualimetrics) and compute cross-validated R2 (Q2) for extracting PCA principal components, and validate the obtained LDA relationships. In order to use the PCA scores as input to LDA, the number of PCs has to be optimized using LOO-CV, and ANOVA should be employed to test whether there are significant differences in PCA scores between the defined groups of Fuzi samples (E1, E2, etc.). Only the significant (with p values < a defined significance level) should be included as input into LDA. Even if PCA-LDA is not employed, ANOVA between the scores of individual groups can be employed for statistical significance testing. Furthermore, PCA contribution plots can be plotted between the clusters to gain direct insight into which input variables are the most influential on their differences (in terms of magnitude, and direction, similar to correlation). The authors should also include performance metrics from the developed supervised classification model (LDA, and/or PCA-LDA) such as accuracy, specificity, sensitivity, area under the ROC curve (10.1016/j.chemolab.2005.05.004) and the Matthews correlation criterion (10.1016/0005-2795(75)90109-9). All these parameters have to be estimated using LOO-CV. Please improve Figure 4. The chromatogram in panel A) is barely legible. This can be solved by separating each chromatogram with an offset. Here, right now it looks as they are already plotted with an offset. If yes, in that case Signal must have an arbitrary unit (a.u.).  Figure 5 also needs to be improved, especially Figure 5D, the points are barely legible. English language needs to be proofed. For instance, "3.6. Date analysis" --> "Data analysis" ?

Author Response

Comment 1: In the Introduction section the authors should further elaborate why RP-HPLC was utilized to analyze Fuzi extracts. Please also make clear (throughout the manuscript) that the separation mode was reversed-phase. A recently published review notes that the RP mode accounts for > 90 % of all HPLC separations in several fields including pharmaceutical chemistry, and analysis of (biologically)-active compounds of plant origin. (10.1021/acs.chemrev.8b00246)

Response: Thanks for the suggestion. In the Introduction section, we have added the reason why we chose RP-HPLC. The reference has also been added.

Comment 2: What was the rationale of choosing Kromasil C18 as the optimal column for the RP-HPLC separation? Were chromatographic tests performed (to determine e.g., hydrophobicity, silanol activity, steric selectivity)? If yes, please elaborate more on the optimization of RP-HPLC conditions in the manuscript. The authors are referred to the previously-mentioned review article for information on RP-HPLC column selection (10.1021/acs.chemrev.8b00246).

Response: Thanks for the comment. We have read this review and obtained many useful information. The chromatographic tests were not performed. However, we screened the frequently-used RP column, including Eclipse Plus C18 column (3.5 μm 100 × 4.6 mm), AlltimaTM C18 column (5 μm 250 × 4.6 mm), and Kromasil C18 column (5 μm 250 × 4.6 mm). The Kromasil C18 column was selected as the optimal column for the RP-HPLC separation.

Comment 3: Instead of performing the PCA&LDA analyses separately, the authors may want to consider coupling PCA (or even PLS) to LDA, whereas the PCA scores (computed from the chromatographic data) would be used as input to LDA in order to calculate the canonical factors instead of chromatographic data. In such a manner the noise levels would be minimized. The models lack comprehensive validation. Since the number of samples is quite low, please employ leave-one-out cross-validation (D. L. Massart, et al. Handbook of Chemometrics and Qualimetrics) and compute cross-validated R2 (Q2) for extracting PCA principal components, and validate the obtained LDA relationships.

Response: Thanks for the helpful suggestion. We have added the cross-validation and the compute cross-validated parameters of PCA model. The R2 and Q2 were 0.817 and 0.563, respectively, which were suitable for fitness and prediction. Figure 6 has been revised.

Comment 4: In order to use the PCA scores as input to LDA, the number of PCs has to be optimized using LOO-CV, and ANOVA should be employed to test whether there are significant differences in PCA scores between the defined groups of Fuzi samples (E1, E2, etc.). Only the significant (with p values < a defined significance level) should be included as input into LDA. Even if PCA-LDA is not employed, ANOVA between the scores of individual groups can be employed for statistical significance testing.

Response: Special thanks for the very helpful suggestion. We have employed ANOVA for statistical significance testing of PCA scores between the defined three groups. PC1 scores of three groups were significant difference between each other (p < 0.01), and PC2 scores of the Fuzi from Jiangyou and the Fuzi from Butuo also showed difference (p < 0.05). Then, the PCA scores were used as input data to LDA in order to calculate the canonical factor. Figure 6C has been revised.

Comment 5: PCA contribution plots can be plotted between the clusters to gain direct insight into which input variables are the most influential on their differences (in terms of magnitude, and direction, similar to correlation).

Response: We have added the PCA contribution plot between the clusters, as shown in Figure 6A.

Comment 6: Please improve Figure 4. The chromatogram in panel A) is barely legible. This can be solved by separating each chromatogram with an offset. Here, right now it looks as they are already plotted with an offset. If yes, in that case Signal must have an arbitrary unit (a.u.). Figure 5 also needs to be improved, especially Figure 5D, the points are barely legible.

Response: We have improved Figure 4 and 5 and changed the signal as an a.u. in Figure 4A.

Comment 7: English language needs to be proofed. For instance, "3.6. Date analysis" --> "Data analysis" ?

Response: We are sorry for the spelling mistake. The language has been proofed again.

Reviewer 3 Report

Discrimination of herbal medicines according to their grown regions is important to determine the quality. This study is interesting and somewhat valuable. However, several things should be considered or corrected.

1. Instead of ‘principal roots’, ‘tap roots’ is more suitable.

2. ‘v/v’ should be an Italic.

3. In ‘2.1. Optimization of the Extraction Conditions’, brief results of optimization process is required to be presented as Supplementary information if possible.

4. In ‘2.2. Optimization of the HPLC Conditions, the reason why the analytical conditions were selected should be written in terms of peak shapes, resolutions, and so on.

5. I think Table 1 is unnecessary as mobile phase ratios are changeable in wide ranges.

6. In Figure 2, statistical analysis should be performed in order to compare the contents of aconitine-related compounds in Fuzi samples from different regions.

7. Like written about monoester-type diterpenoid alkaloids, the contents of diester-type diterpenoid alkaloids is needed to be described individually.

8. In Figure 2, Simply the contents of acontine-related compounds cannot guarantee the medicinal quality of Fuzi. Macroscopic, pharmacological, and therapeutic evidence is strongly required for evaluation of the quality of herbal medicines.

9. In Figure 3, It does not make sense why mesaconitine was the criteria for comparison with other compounds. Its highest content is not a rationale for the comparison. Rather, the ratio of diester-type/monoester type is more reasonable.

10. In Figure 4, is there any reason why peaks for fingerprinting does not include entire marker compounds?

11. In Figure 5B, Similarity point included was observed. What does it mean?

12. Figures in Figure 5 is needed to be re-arranged by same size or by the statistical results.

Author Response

Comment 1: Instead of ‘principal roots’, ‘tap roots’ is more suitable.

Response: The word has been revised.

Comment 2: “v/v” should be an Italic.

Response: The symbols have been revised.

Comment 3: In ‘2.1. Optimization of the Extraction Conditions’, brief results of optimization process is required to be presented as Supplementary information if possible.

Response: The “Optimization Process of the Extraction Conditions” has been added in the Supplementary Material. In addition, “Six representative alkaloids contents of Fuzi from Jiangyou, Butuo, and Yunnan (Table S1)” and “Similarity results of HPLC fingerprint of Fuzi from Jiangyou, Butuo, and Yunnan (Table S2)” have been added.

Comment 4: In ‘2.2. Optimization of the HPLC Conditions, the reason why the analytical conditions were selected should be written in terms of peak shapes, resolutions, and so on.

Response: We have added the description of evaluation indicators in section 2.2.

Comment 5: I think Table 1 is unnecessary as mobile phase ratios are changeable in wide ranges.

Response: Thank you for your suggestion. Eluting gradient examination is a part of the optimization of the HPLC conditions. To clearly display the specific examined mobile phase ratios, we want to remain the table.

Comment 6: In Figure 2, statistical analysis should be performed in order to compare the contents of aconitine-related compounds in Fuzi samples from different regions.

Response: Thanks for the good comment. Statistical analysis has been performed, and Figure 2 has been revised.

Comment 7: Like written about monoester-type diterpenoid alkaloids, the contents of diester-type diterpenoid alkaloids is needed to be described individually.

Response: We have added the description in section 2.4 according to the suggestion.

Comment 8: In Figure 2, Simply the contents of acontine-related compounds cannot guarantee the medicinal quality of Fuzi. Macroscopic, pharmacological, and therapeutic evidence is strongly required for evaluation of the quality of herbal medicines.

Response: Special thanks to the reviewer for the good comment. Indeed, the contents of acontine-related compounds and HPLC fingerprint are only two chemical methods for evaluating the quality of Fuzi. Other macroscopic, pharmacological, or therapeutic evidence is required for comprehensive evaluation of the quality. In this study, we have found an interesting result that the contents of aconitine-related compounds have a strong connection with the geographical origins of Fuzi. We want to perform macroscopic analysis and pharmacological studies to investigate the difference of Fuzi from different geographical origins in the follow-up study.

Comment 9: In Figure 3, It does not make sense why mesaconitine was the criteria for comparison with other compounds. Its highest content is not a rationale for the comparison. Rather, the ratio of diester-type/monoester type is more reasonable.

Response: Because benzoylhypaconine and benzoylaconine have very low contents or could not be quantified, we studied ratios of the other four alkaloids. It was found that the ratios of mesaconitine to the other three alkaloids (benzoylmesaconine/mesaconitine, hypaconitine/mesaconitine, and aconitine/mesaconitine) were most effective in differentiating the origin of Fuzi. Particularly, the content ratio of benzoylmesaconine/mesaconitine may be a chemical feature that can be used to differentiate the origin of Fuzi.

According to the good suggestion, we analyzed the ratio of diester-type/monoester-type alkaloids in all Fuzi batches. The result also indicated the differences of Fuzi from three geographical origins. Therefore, we added the ratio of diester-type/monoester-type alkaloids as another chemical feature to differentiate the origin of Fuzi. Thanks again for the good comment.

Comment 10: In Figure 4, is there any reason why peaks for fingerprinting does not include entire marker compounds?

Response: The multicomponent quantification showed that the contents of benzoylhypaconine and benzoylaconine were very low or could not be quantified or detected in most Fuzi samples, especially the Fuzi from Butuo and Yunnan (Figure 2). Thus, the HPLC fingerprint does not include benzoylhypaconine and benzoylaconine.

Comment 11: In Figure 5B, Similarity point included was observed. What does it mean?

Response: In addition to the contents of 6 representative alkaloids, we also used the fingerprint similarity to form a 7 × 32 data matrix to conduct data analysis. All PCA contribution plots were showed in Figure 5B.

Comment 12: Figures in Figure 5 is needed to be re-arranged by same size or by the statistical results.

Response: Figures 5 and 6 has been revised according to the suggestions of reviewers 2 and 3.

Round 2

Reviewer 1 Report

In my opinion the lack of novelty is in this work. Very similar paper was published in 2018 (DOI: 10.1002/jssc.201800937), where the same plant, standards and general concept of work were presented. However, after reading the authors answers I am willing to accept this manuscript.  If the evaluation of other Reviewers and Editors is positive, I also agree with this decision.

Author Response

Comment: In my opinion the lack of novelty is in this work. Very similar paper was published in 2018 (DOI: 10.1002/jssc.201800937), where the same plant, standards and general concept of work were presented. However, after reading the authors answers I am willing to accept this manuscript. If the evaluation of other Reviewers and Editors is positive, I also agree with this decision.

Response: Thank you for accepting our responses and the revised manuscript.

Reviewer 2 Report

The authors have notably improved the manuscript. However, there are still a few minor issues.

1) Figure 4 (chromatograms) still needs to be improved. In Figure 4A since the response is in an arbitrary unit, the tick labels (0-400) should be removed. Both A) and B) should be considerably improved. The figures are of a really low resolution. The text is too small and cannot be read.

2) In the Figure 6 caption: A) is the PCA bi-plot (of loadings & scores), but B) is only the PCA score plot. What is the reason the authors chose to show both?

3) Upon using the PCA scores (instead of raw chromatographic data) the cross-validated classification accuracy decreased from ~93.8 % to 90.6 %. How would the authors comment on this? Please also report cross-validated specificity, and sensitivity values either using the one-against-one or one-against-all strategy.

4) P11L333: "or SIMCA-P" --> "and SIMCA-P". Please comprehensively proof the spelling throghout the manuscript.

5) Why was the fingerprint similarity of the 32 batches included as a variable in the multivariate analyses and the classification? This has not been clarified by the authors.

Author Response

Comment 1: The authors have notably improved the manuscript. However, there are still a few minor issues. Figure 4 (chromatograms) still needs to be improved. In Figure 4A since the response is in an arbitrary unit, the tick labels (0-400) should be removed. Both A) and B) should be considerably improved. The figures are of a really low resolution. The text is too small and cannot be read.

Response: Thanks for the helpful comment. We have removed the tick label, enlarged the text, and improved the resolution (400 dpi) of Figure 4.

Comment 2: In the Figure 6 caption: A) is the PCA bi-plot (of loadings & scores), but B) is only the PCA score plot. What is the reason the authors chose to show both?

Response: Figure 6A is the PCA bi-plot (of loadings & scores). It also displays the correlation of original variables to principle components and samples. Figure 6B only shows the PCA score plot and clearly displays the clustered results in different color. Therefore, we showed both the figures of biplot and score plot.

Comment 3: Upon using the PCA scores (instead of raw chromatographic data) the cross-validated classification accuracy decreased from ~93.8 % to 90.6 %. How would the authors comment on this? Please also report cross-validated specificity, and sensitivity values either using the one-against-one or one-against-all strategy.

Response: According to the comment in the first round of review, we used the PCA scores to replace the raw data of alkaloid contents. Although the accuracy decreased from 93.8 % to 90.6 %, the noise levels of the analyses were minimized. Performing LDA based on PCA is found to reduce the chance of over-fitting that may occur with a pure LDA model. We think that the classification model was improved.

The specificity values and the sensitivity values have been added in section 2.8.

Comment 4: P11L333: "or SIMCA-P" --> "and SIMCA-P". Please comprehensively proof the spelling throughout the manuscript.

Response: ANOVA, HCA, and LDA were analyzed using SPSS 20.0 (SPSS Statistics, Chicago, IL, USA). PCA were analyzed using SIMCA-P 13.0 (Umetrics, Umea, Sweden). We have revised the description.

Comment 5: Why was the fingerprint similarity of the 32 batches included as a variable in the multivariate analyses and the classification? This has not been clarified by the authors.

Response: The similarity values of samples are relative values that are calculated against the reference fingerprint generated with the 32 chromatograms of Fuzi. The similarity values may be changed when another reference fingerprint is used. Thus, we agree with the comments of reviewers 2 and 3 (The 'Similarity' is not the marker compound in this experiment). In this revised manuscript, we have deleted the variable and used 6 marker compounds forming the data matrix to conduct the analyses. The Figures and the descriptions have been revised.

Reviewer 3 Report

In Figure 6A (revised version), The question I mentioned before is not suitably answered. PC plot contains 7 loading plot point(6 marker compounds + Similarity). Usually loading plot is made by marker compounds analyzed in the matrix (marker compounds x samples). But, the 'Similarity' is not the marker compound in this experiment. Please explain what 'Similarity' means in the PC plot.

Author Response

Comment: In Figure 6A (revised version), The question I mentioned before is not suitably answered. PC plot contains 7 loading plot point (6 marker compounds + Similarity). Usually loading plot is made by marker compounds analyzed in the matrix (marker compounds x samples). But, the 'Similarity' is not the marker compound in this experiment. Please explain what 'Similarity' means in the PC plot.

Response: The similarity values of samples are relative values that are calculated against the reference fingerprint generated with the 32 chromatograms of Fuzi. The similarity values may be changed when another reference fingerprint is used. Thus, we agree with the comments of reviewers 2 and 3 (The 'Similarity' is not the marker compound in this experiment). In this revised manuscript, we have deleted the variable and used 6 marker compounds forming the data matrix to conduct the analyses. The Figures and the descriptions have been revised.